

# Storylines of extreme summer temperatures in southern South America

Solange Suli[1,2,3], David Barriopedro[1], Ricardo García-Herrera[1,2], Soledad Collazo[1,2,3], Antonello Squintu[4], Matilde Rusticucci[3]

[1]Institute of Geosciences (IGEO), Spanish National Research Council – Complutense University of Madrid (CSIC-UCM), Madrid Spain
[2]Complutense University of Madrid, Faculty of Physical Sciences, Physics of the Earth and Astrophysics, Madrid, Spain
[3]University of Buenos Aires, Faculty of Exact and Natural Sciences, Department of Atmospheric and Ocean Sciences, Buenos Aires, Argentina
[4]CMCC Foundation - Euro-Mediterranean Center on Climate Change, Bologna, Italy

*Correspondence to*: Solange Suli (ssuli@ucm.es)

**Abstract.** Understanding the sources of uncertainty in future climate extremes is crucial for developing effective regional adaptation strategies. This study examines projections of summer absolute maximum temperature (TXx) over four regions of southern South America: northern, central-eastern, central Argentina, and southern areas. We analyse simulations from 26 global climate models and apply a storyline approach to explore how different climate drivers combine to shape future changes in TXx for the late 21st century (2070–2099).

The storylines are based on changes in key physical drivers, including mid-tropospheric circulation, regional soil moisture, sea surface temperature in Niño 3.4 region, and the intensity of the South Atlantic Convergence Zone. A multi-linear regression framework reveals that the dominant drivers of the projected warming in TXx vary substantially across regions. In northern areas, warming is primarily influenced by remote drivers such as tropical sea surface temperatures and changes in the South Atlantic Convergence Zone. Central-eastern and central Argentina exhibit mixed local and remote influences, while southern regions are predominantly affected by local drivers (soil drying and atmospheric circulation changes).

Together, these drivers explain up to 56% of the inter-model spread in future projections of TXx. However, their ability to account for the uncertainty in percentile-based indices and regional heatwave characteristics is more limited, suggesting that complex heat metrics may be influenced by additional processes.

## 1 Introduction

Global mean surface temperature has been approximately 1.1°C higher in the 2011–2020 than 1850–1900, with larger increases over land than over the oceans (IPCC, 2023). As a result of this warming, significant negative impacts have already been observed across various sectors of the society, including e.g. risks in water and food security (e.g., El Bilali et al., 2020; Stringer et al., 2021) or severe health effects driven by the increasing frequency of heat waves (HWs; e.g., Amengual et al., 2014; Anderson and Bell, 2009; Ballester et al., 2023; Chesini et al., 2022). While it is unequivocal that human influence has



contributed to atmospheric warming, its manifestations and impacts vary across different regions. Particularly, in South America (SA), the Sixth Assessment Report (AR6) of the Intergovernmental Panel on Climate Change (IPCC) indicates that near-surface temperatures have been increasing over the past several decades, but with pronounced regional variations (IPCC,
2023). For instance, southwestern SA, particularly the Andean region, has experienced an outstanding warming (e.g., Suli et al., 2023; Vuille et al., 2015), with temperatures rising faster than the global average (IPCC, 2021). Likewise, observed trends in temperature extremes are uneven across the SA region. Northern SA reports the strongest trend in the number of days exceeding the 90th percentile during 1950–2018 (Dunn et al., 2021). However, central-southeastern SA shows contrasting results, with some studies reporting decreasing trends in warm extremes (e.g., TXx and TX90) during the austral summer
(Rusticucci et al., 2017; Skansi et al., 2013; Wu and Polvani, 2017), and others indicating significant increases in the frequency of warm season HW days over central Argentina (Suli et al., 2023). Finally, in the southernmost part of SA, there is insufficient evidence to determine clear trends in hot extremes due to limited data availability (IPCC, 2023).

Global Climate Models (GCMs) from the Coupled Model Intercomparison Project (CMIP) have been widely used as the main tool to assess future changes in the mean and extreme values at global and continental scales (Almazroui et al., 2021b; Tebaldi
et al., 2021). In SA, there are several studies on climate change projections (Almazroui et al., 2021a; Bustos Usta et al., 2022 and references therein; Feron et al., 2019; Gulizia et al., 2022; Ortega et al., 2021; Salazar et al., 2024). For instance, Almazroui et al. (2021a) evidence a substantial warming across SA, with annual mean temperature increases ranging from 2.8°C to over 5.0°C under the high-emission scenario SSP5-8.5 by the end of the century (2080–2099). The strongest warming is expected in tropical regions, particularly in the Amazon and at high altitudes such as the Andes. The latter has also been identified as a
hotspot by Salazar et al. (2024), who suggest that amplified warming in the Andes may be linked to elevation-dependent responses. In southern SA, Almazroui et al. (2021a) report a weaker warming (~3°C) than in other regions, which contrasts with North America, where higher latitudes tend to exhibit stronger warming signals (Almazroui et al., 2021c). In spite of this, for 3°C global warming levels, southeastern SA could experience a ~25% increase in warm days (TX90) compared to the 1981–2000 period (Gulizia et al., 2022).

Uncertainties in GCM projections evidenced in the multi-model ensemble cannot be directly interpreted in a probabilistic sense (Shepherd, 2019). To address structural uncertainties, Zappa and Shepherd (2017) propose a storyline-based approach, which provides physically coherent representations of plausible changes at regional scale. Each storyline is constructed by combining climate change responses based on well-known drivers that characterise the regional climate. The combination of storylines manages to capture the range of uncertainty in the future projections from multi-model ensembles (Zappa, 2019). This
methodology has been applied in various regions worldwide (e.g., Bjarke et al., 2024; Gibson et al., 2024; Mindlin et al., 2020; Schmidt and Grise, 2021; Zappa and Shepherd, 2017), focusing mainly on atmospheric circulation patterns and their impacts on precipitation and droughts. Moreover, Garrido-Perez et al. (2024) extend its application to explore the uncertainty of future summer warming over Iberia. However, to date, this approach has not been used to analyse uncertainties in the response of SA summer temperature extremes.



Various studies have demonstrated the influence of both local and remote forcings on temperature extremes in SA (Cai et al., 2020; Reboita et al., 2021; Rusticucci et al., 2003). In particular, midlatitudes of SA are strongly influenced by large-scale extratropical circulation patterns, such as waveguides, which often cause enhanced ridging activity over southern SA (O'Kane et al., 2016). Rossby wave trains also favour the strengthening of the subtropical jet over SA, increase the advection of cyclonic vorticity over southeastern SA and transport warm and moist air from the north into this region (Grimm and Ambrizzi, 2009).

Likewise, Rossby wave activity is closely linked to the El Niño-Southern Oscillation (ENSO), one of the primary modes of interannual variability affecting SA (Barreiro, 2010; Cai et al., 2020; Fernandes and Grimm, 2023; Grimm and Tedeschi, 2009; Reboita et al., 2021; Rusticucci and Kousky, 2002). Most studies about ENSO impacts over SA have focused on precipitation, while its influence on summer extreme temperatures remains less explored. Although the strongest ENSO-related temperature signals in southern SA have been documented during austral winter (Cai et al., 2020; Müller et al., 2000), Rusticucci et al.

(2017) reported that El Niño events are associated with a reduced diurnal temperature range north of 40°S in austral summer, suggesting a modulation of extreme temperatures during summer as well (Mc Gregor et al., 2022).

In addition, the South Atlantic Convergence Zone (SACZ) represents an important climatological feature of the austral summer in SA (Barros et al., 2000; Carvalho et al., 2003; Collazo et al., 2024). Particularly, an active SACZ promotes subsidence conditions over southeastern SA, favouring the development of an anticyclonic circulation there, which in turn causes warming

particularly given the relatively dry conditions of the warm season (Cerne and Vera, 2011). Moreover, changes in the position or intensity of the SACZ are expected to affect HWs over the region. In this context, Zilli et al. (2019) and Zilli and Carvalho (2021) have identified a poleward shift of the SACZ in response to climate change, based on satellite-gauge precipitation data and GCM simulations. However, the disagreement among GCMs and ensemble members on simulated precipitation changes introduces substantial uncertainty in future projections of the SACZ

(Carvalho and Jones, 2013).

The uncertainty associated with changes in thermodynamic components such as temperature is also modulated by non-dynamical drivers like soil-moisture coupling (Cheng et al., 2017; Hsu and Dirmeyer, 2023; Ma and Xie, 2013; Trugman et al., 2018; Vogel et al., 2017; Zhou et al., 2024). SA has been identified as a key hotspot for land–atmosphere interactions (Sörensson and Menéndez, 2011; Spennemann et al., 2018), where soil-moisture plays a crucial role in modulating surface air

temperature variability (Coronato et al., 2020; Guillevic et al., 2002; Menéndez et al., 2019; Seneviratne et al., 2010). Ruscica et al. (2016) found a strong land-atmosphere coupling in central Argentina during the summer for both present and future climates. However, in northern Argentina, Uruguay, and southern Brazil, this interaction was projected to weaken in the future. These findings underscore the complexity of assessing future projections of temperature extremes due to the multiplicity and heterogeneity of drivers across SA regions. To address this challenge, this study employs a storyline approach to dissect the

climate change responses of maximum summer temperature in four regions of southern SA (SSA), aiming to better understand the drivers of structural uncertainties in GCM projections. This approach reconstructs regional projections and their associated uncertainties based on changes in different drivers of the regional climate change (Garrido-Perez et al., 2024; Mindlin et al., 2020; Zappa and Shepherd, 2017).



This paper is structured as follows: Sect. 2 describes the datasets and methodology used to identify key drivers for each SSA
region and to construct the storylines. Sect. 3 presents the results, including the projected changes in the drivers, the sensitivity
of the maximum summer temperature changes to these drivers, and a quantitative analysis of SSA summer temperature
responses obtained from the storylines. Finally, the main findings are summarised and discussed in Sect. 4.

## 2 Data and methodology

### 2.1 Data

We used daily maximum temperature at 2 meters (T2m) from the ERA5 reanalysis over SSA ([25, 60] °S and [80, 40] °W)
with a regular 2.5° resolution during the austral warm seasons (Oct–Mar) of 1979–2023 (Hersbach et al., 2020). We also
employed data from 26 GCMs of the Climate Model Intercomparison Project Phase 6 (CMIP6, see Table S1 for details). Daily
maximum near-surface (2 meters) air temperature (TX) was used for the definition of extreme temperature indices. In addition,
monthly fields of sea surface temperature (SST), soil moisture content (SM, within the top 0–10 cm of the soil), 500 hPa
geopotential height (Z500) and outgoing longwave radiation (OLR) were employed for the construction of the drivers (see
Sect. 2.3). GCM historical simulations (Eyring et al., 2016) over the period 1979–2014 and Shared Socioeconomic Pathway
projections (SSP5-8.5, O'Neill et al., 2016) for the 2015–2099 period were obtained from the CMIP6 archive. A common 2.5°
x 2.5° horizontal grid and the austral summer season (December-January-February, DJF) were considered for both reanalysis
and GCM simulations. Bilinear interpolation was used for most of the variables, while a conservative remapping was applied
to SM data to avoid spurious values (Jones, 1999).

For most of the analyses, extreme temperature conditions are diagnosed based on the absolute summer maximum of TX (TXx).
This index emphasizes the magnitude of extreme events, rather than their frequency or duration, assuming that extremes occur
every summer. TXx is computed at each grid point, and at regional scales, using the regions defined in the next section.
Regional TXx was calculated by first averaging TX over the region and then selecting the absolute maximum value for each
summer in order to ensure warm widespread conditions at the regional level.

### 2.2 Regionalisation

To identify spatially coherent regions, we followed the clustering procedure of Suli et al. (2023) for weather stations. Herein,
the identification of homogeneous regions is based on clustering grid points with a high co-occurrence of local HW days. To
do so, we defined HW events at each grid point as sequences of at least three consecutive days in which T2m exceeded the
local daily 90th percentile (Pc90) of the 1981–2010 baseline period, using a 31-day moving window. All days comprising a
HW event are classified as local HW days. Then, we applied the bottom-up Ward's hierarchical clustering method (Ward,
1963) to identify land grid points with a high co-occurrence of HW days (see Sect. 2.2 of Suli et al. (2023) for further details).
As a result, five climatologically homogeneous regions were identified in SSA, which are consistent with those obtained from
station-based data in Suli et al. (2023). The identified regions are depicted in Figure 1, and named as northern SSA (NS),





central-eastern SSA (CES), central Argentina and northern Argentinian Patagonia (CA), central Chile (CCH), and southern

SSA (SS), including Argentinian Patagonia and southern Chile.

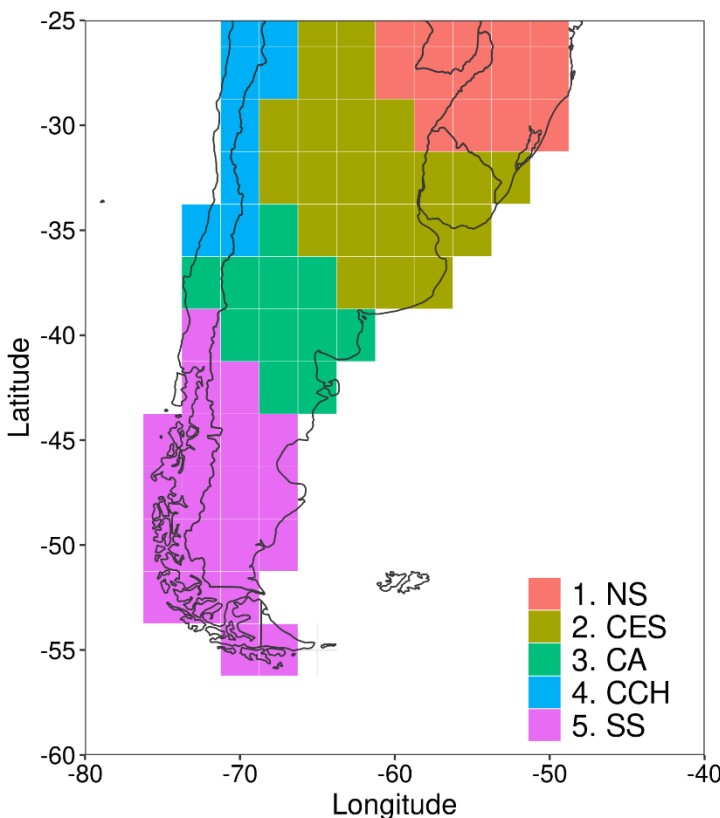

**Figure 1: Regionalisation of SSA based on the co-occurrence of HW days during the warm seasons of 1979–2023. Grid points are coloured and numbered from 1 to 5, according to the region they belong: C1—northern of SSA (NS), C2—central-eastern of SSA**
**(CES), C3—central Argentina and northern Argentinian Patagonia (CA), C4—central Chile (CCH), C5—Argentinian Patagonia and southern Chile, southern SSA (SS).**

To ensure consistency in the spatial analysis, the same SA regionalisation of Figure 1 was also applied to each CMIP6 GCM.

However, the CCH was excluded from the analysis due to substantial temperature biases associated with unresolved

topography in GCMs, which can reach magnitudes of up to ~8°C in northern Chile (Salazar et al., 2024).

**2.3 Definition of drivers**

For the regional analysis, the following local and remote drivers were considered (for more details on the candidate drivers,

see the Introduction section, and references therein):

- Sea surface temperature in Niño 3.4 region (N3.4): mean summer SST in the Niño 3.4 region ([5°N–5°S; 120°–170°W]).





•   SSA geopotential height (Z500): mean summer Z500* averaged over the [40°–55°S, 60°–80°W] domain
       (see green box in Fig. 3h), with Z500* being the departure of the Z500 from its zonal mean.

    •   Regional soil moisture ($SM_i$): mean summer SM, averaged over the region i, with i being one of the SSA
       regions (SS, CA, CES or NS). We also tested the performance of drivers extending across more than one
       SSA region, and selected consequently a northern Argentinian subregion ($SM_{north}$, [21°–31°S, 54°–66°W]).

•   Intensity of the SACZ (SACZ): difference in summer mean OLR between two domains spanning 10° of
       latitude and 15° of longitude ([33°–43°S, 30°–45°W] and [25°–35°S, 10°–25°W]), as depicted in Figs. 3b, d
       and f).

**2.4 Storyline methodology**

For each region, storylines describe the combined effect of the drivers' changes on summer TXx projections. Climate change
responses, denoted as Δ, are computed for TXx and the drivers as the difference of the summer mean between the far future
(2070–2099) and the historical period (1979–2014). The methodology follows the framework proposed by Zappa and Shepherd
(2017) and is briefly described below.

Firstly, we computed the climate change responses of the drivers for each region and each GCM. Secondly, the regional ΔTXx
response was modelled separately for each region using an ordinary multi-linear regression (MLR, Eq. 1). In this regression,
ΔTXx is the dependent variable (target), while two drivers act as independent variables (predictors):

$$\frac{\Delta TXx}{GW} = a_x + b_x * \left(\frac{\Delta D_1}{GW}\right)'_m + c_x * \left(\frac{\Delta D_2}{GW}\right)'_m \tag{1}$$

We only considered two drivers per region in order to limit the number of storylines (given by $2^n$, with n being the number of
drivers). The selection of two drivers also enables to physically interpret the storylines, and helps to avoid overfitting in the
MLR caused by interdependencies among the predictors. In Eq. (1), $\Delta D_1$ and $\Delta D_2$ represent the changes in the two drivers for
each model m. The symbol (') indicates the standardised change relative to the multi-model mean (MMM), $a_x$, $b_x$ and $c_x$ are
the regression coefficients: $a_x$ denotes the MMM intercept, representing the expected mean response when there is no
deviation in the driver responses relative from the MMM; $b_x$ and $c_x$ quantify the sensitivity of regional ΔTXx to each driver.
Both the target (ΔTXx) and the drivers ($\Delta D_1$ and $\Delta D_2$) were scaled by global warming (GW) defined as the corresponding
change in the area-weighted global mean near-surface temperature. The MLR is based on 26 values (GCMs) and was computed
separately for each region.

Once the sensitivity coefficients were obtained, regional ΔTXx can be estimated for given values of GW and drivers' responses.
Combining opposite (strong or weak) responses of the two drivers for each region results in four different storylines, which
reflect the corresponding effects in ΔTXx. The final ΔTXx response follows Eq. (2):

$$\frac{\Delta TXx}{GW} = a_x + b_x * t + c_x * t \tag{2}$$



Here, *t* denotes the storyline index, which measures the magnitude of the driver responses (in standard deviations). In this case, the changes of the two drivers were selected to have equal standardised amplitudes, which also allows us comparing their relative effects in ΔTXx. As described by Zappa and Shepherd (2017), *t* was chosen to lie within the 80% confidence region of the drivers' responses (see black stars in Fig. 4), which was obtained by fitting a bivariate normal distribution ($t\sim \pm 1.26$ std). Full details of the methodology can be found in Zappa and Shepherd (2017), in the Appendix A of Mindlin et al. (2020)

and in Garrido-Perez et al. (2024).

## 3 Results

### 3.1 Variability of projected changes

Figure 2 shows the MMM summer projections of TXx and the drivers used in this study for 2070–2099 (with respect to 1979–2014). Consistent with Almazroui et al. (2021a), tropical regions in northern SA exhibit a strong significant warming by the

end of the century, exceeding 5°C. In SSA, the largest TXx increases are projected along the Andes Mountains (Fig. 2a), aligning with Salazar et al. (2024), who reported a warming of up to 6°C in northern Chile. In contrast, central SSA regions display a more homogeneous and less pronounced warming of approximately 4℃ (Lagos-Zúñiga et al., 2024).

Concerning the projected changes in the climate drivers, SM is expected to decrease significantly over northern SA, especially in the Amazon and along the Andes Mountains (see Fig. 2b), consistent with Cheng et al. (2017). In contrast, future SM

projections for central-eastern SSA remain uncertain. In this region, CMIP5 GCMs projected positive SM changes for 2061–2080 compared to 2006–2025 under a high emission scenario (Cheng et al., 2017). However, more recent CMIP6 projections under SSP5-8.5 show no consistent changes in regional SM (Cook et al., 2020). As this region is characterised by strong soil-atmosphere coupling, the uncertainties in SM projections are expected to propagate to summer temperature changes. Regarding SSTs, the central-eastern Pacific is projected to warm by up to 4°C above historical values by the end of the century (Fig. 2c).

This warming enhances convection over the region as it can be seen by negative changes in OLR (Fig. 2d). Pronounced warming is also observed in the western Pacific Ocean near southeastern Australia (Fig. 2c), as noted by other authors (Lenton et al., 2015; Oliver et al., 2014). Although the direct impact of the western Pacific Ocean warming on SA remains uncertain, Sun et al. (2023) suggested that air-sea coupling in the tropical Pacific greatly amplifies the atmospheric response of the South Pacific to ENSO. Indeed, ΔZ500 exhibits alternating anomalies over the Pacific that resemble a Rossby wave pattern extending

from southern Australia (Fig. 2e), which has been associated with HWs in the subtropical SA (Cerne and Vera, 2011; Shimizu and de Cavalcanti, 2011). In addition, enhanced anticyclonic conditions are projected in southern SA, particularly at high latitudes, which may be linked to an increasing zonal asymmetry of the Southern Annular Mode during DJF (Campitelli et al., 2022).





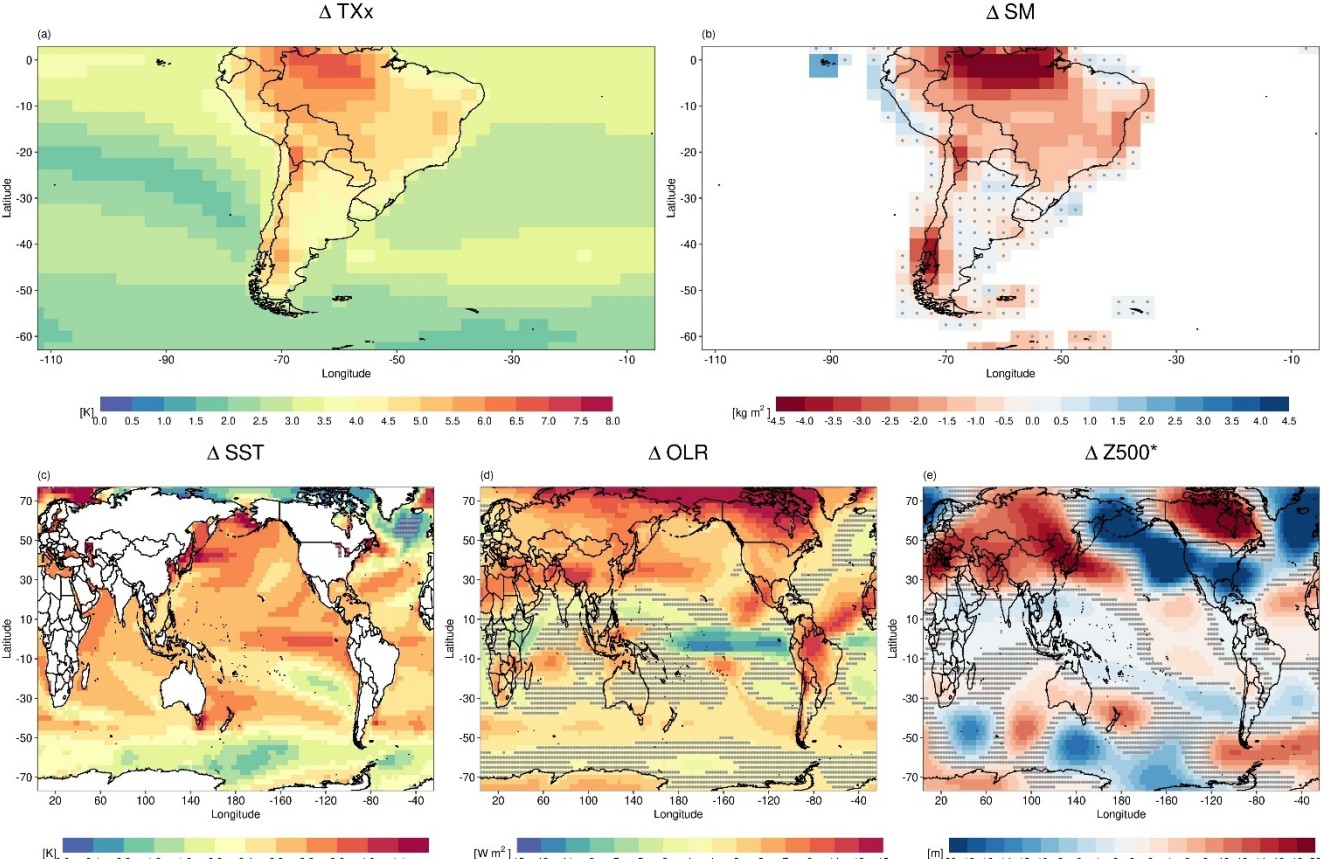

**Figure 2: Multi-model mean (MMM) summer (DJF) changes in (a) Absolute Maximum Temperature (TXx, K), (b) Soil Moisture (SM, kg m⁻²), (c) Sea Surface Temperature (SST, K), (d) Outgoing Longwave Radiation (OLR, W m⁻²) and (e) Geopotential Height at 500 hPa with the zonal mean removed (Z500\*, m). Changes are computed as the difference between the periods 2070–2099 and 1979–2014. Grey dots indicate areas where changes are not statistically significant at the 95% confidence level, based on a two-tailed t-test.**

## 3.2 Selection of drivers

We first perform different MLR models (see Eq. 1) for each SSA region, testing different combinations of local and remote climate drivers. Sensitivity tests were also conducted to assess whether lagged relationships between the drivers and regional TXx could improve the model performance. However, no significant improvement was found when introducing temporal lags. Therefore, we focused on simultaneous summer changes in both drivers and ΔTXx. Based on these analyses, the two drivers with the lowest p-value and physically consistent influences on ΔTXx were selected for each region. To ensure the robustness of the analysis, we verified that the selected drivers were not significantly correlated with each other (i.e. Pearson correlation coefficients with p-values > 0.1).

The final combination of drivers is outlined in Table 1, along with the sign of their regression coefficients (+/-) and the corresponding explained variance ($R^2$). The climate drivers vary across SSA regions. In NS, the warming response in TXx is



substantially affected by changes in remote drivers ($\Delta$N3.4 and $\Delta$SACZ), while in SS only local drivers are identified ($\Delta$SM and $\Delta$Z500). In contrast, both local and remote drivers ($\Delta$SM and $\Delta$SACZ) affect the warming of extremes in CES and CA. For all regions, the uncertainty in the drivers' changes is significantly correlated with that in TXx, except for CES, where $\Delta$SM$_{north}$ does not show a significant response in $\Delta$TXx. Although this region exhibits a strong soil-atmosphere coupling on interannual timescales (Jung et al., 2010; Ruscica et al., 2015; Sörensson and Menéndez, 2011), the lack of significance indicates that SM cannot explain the spread of TXx projections in this area. For all regions, $R^2$ exceeds 35%, with the highest values observed in SS ($R^2\sim 41\%$) and CA ($R^2\sim 56\%$).

| Region | $D_1$ | $D_2$ | $R^2$ |
|--------|-------|-------|-------|
| NS | **$\Delta$N3.4 (+)** | **$\Delta$SACZ (+)** | 0.37 |
| CES | $\Delta$SM$_{north}$ (-) | **$\Delta$SACZ (+)** | 0.35 |
| CA | **$\Delta$SM$_{CA}$ (-)** | **$\Delta$SACZ (+)** | 0.56 |
| SS | **$\Delta$SM$_{SS}$ (-)** | **$\Delta$Z500 (+)** | 0.41 |

**Table 1: Drivers used to perform the MLR (Eq. 1) for each SSA region. The symbol in parentheses ('+' or '–') specifies the sign of the regression coefficient and bold values denote statistically significant coefficients (p < 0.1). The last column indicates the $R^2$ obtained from the MLR for each SSA region.**

To further support the choice of the selected regional drivers, we examined the spatial patterns of drivers' influence on regional $\Delta$TXx. Figure 3 illustrates the corresponding sensitivity patterns, as obtained from a MLR of regional $\Delta$TXx onto grid-point drivers' changes. Figures 3 a-b indicate that remote drivers exhibit a significant influence on the $\Delta$TXx response of NS. Significant positive SST regression coefficients over the tropical Pacific (Fig. 3a) suggest that enhanced El Niño events contribute to an increase in NS TXx. Although El Niño is currently associated with cooler TXx conditions in this region compared to La Niña (Arblaster and Alexander, 2012), future projections suggest a weakening of the ENSO-related temperature signal over SSA (Mc Gregor et al., 2022). Consequently, El Niño events may exert a reduced cooling effect in the future, resulting in higher TXx values relative to the present, and thus contributing to a positive $\Delta$TXx response. In addition, a poleward displacement of the SACZ is also associated with amplified warming over this region, as evidenced by negative (positive) $\Delta$OLR values over the poleward (equatorward) box of Fig. 3b. The SACZ, which typically extends south-eastward from the Amazon Basin into the Atlantic Ocean (Liebmann et al., 2004), has also shifted polewards in recent years (Zilli et al., 2019). Similarly, $\Delta$SACZ also acts as a remote driver of $\Delta$TXx in CES. Enhanced convection over the SACZ, associated with a significant intensification of the OLR gradient in Figures 3b and d, reinforces subsidence and strengthens the anticyclonic circulation over northeastern Argentina, Brazil, and northern Uruguay (Suli et al., 2023). Furthermore, drying over northern Argentina and Paraguay (green box in Fig. 3c) is consistent with a warming response in CES. However, this driver does not exhibit a statistically significant signal, possibly reflecting a weakened soil–atmosphere coupling under future climate conditions (Ruscica et al., 2016). Regarding CA (Figs. 3 e-f), the results show that GCMs with larger decreases in SM$_{CA}$ or more pronounced poleward shifts of the SACZ display exacerbated warming of TXx. Finally, the largest warming in SS (Figures 3g-h) is associated with a decrease in SM$_{SS}$ content and an anomalously high Z500 over SS. This is consistent with





(Collazo et al., 2024), who found that during the warm season, southern SA exhibits a strong soil-atmosphere coupling, despite
the region's aridity. Additionally, several studies have shown that anticyclonic anomalies over southern SA are often embedded
in a large-scale Rossby wave train pattern, which can trigger HWs in the region (Cerne and Vera, 2011; Jacques-Coper et al.,
2016). In the following, the changes in the selected regional drivers (boxes in Fig. 3) will be employed to construct the
storylines of future changes in regional ΔTXx.





**Figure 3: Sensitivities of summer changes in absolute maximum temperature (ΔTxx, 2070–2099 minus 1979–2014) associated with uncertainties in the responses of key drivers for each region determined using a multi linear regression model (see Eq. 1): NS: (a) ΔN3.4 and (b) ΔSACZ; CES: (c) ΔSM and (d) ΔSACZ; CA: (e) ΔSM and (f) ΔSACZ and SS: (g) ΔSM and (h) ΔZ500—. In regions where only one driver was available, a simple linear regression was applied. Colours represent ΔTXx responses scaled by global warming (GW, K) and standardised with respect to the multi-model mean (MMM). The boxes highlight the domains employed for**
**the construction of regional drivers. Local (non-local) drivers are denoted in green (yellow). Stippling areas denote statistically significant regression coefficients at p < 0.1, after a two-tailed t-test.**

### 3.3 Storylines analysis

Four storylines (herein labelled as ST#, with # ranging from 1 to 4) of future summer TXx changes were constructed based on the combination of the two most influential drivers of ΔTXx in each region described in Sect. 3.2. Figure 4 depicts the
scatterplots of the two drivers' responses within the CMIP6 ensemble, along with the standardised change amplitudes selected to construct each storyline (represented by black stars), following the regression framework described in Sect. 2.3. GCMs that displayed a systematic outlier behaviour across all regions were excluded from the analysis. The results show considerable uncertainty. Some drivers show consistent changes in sign but have uncertain magnitudes (e.g., SM in SS and the N3.4 index), while for others, both the sign and magnitude of the change are uncertain (e.g., SACZ or SM in CA). Each storyline
characterises the summer ΔTXx as the result of combined responses in $b_x$ and $c_x$ (see Eq. 2).

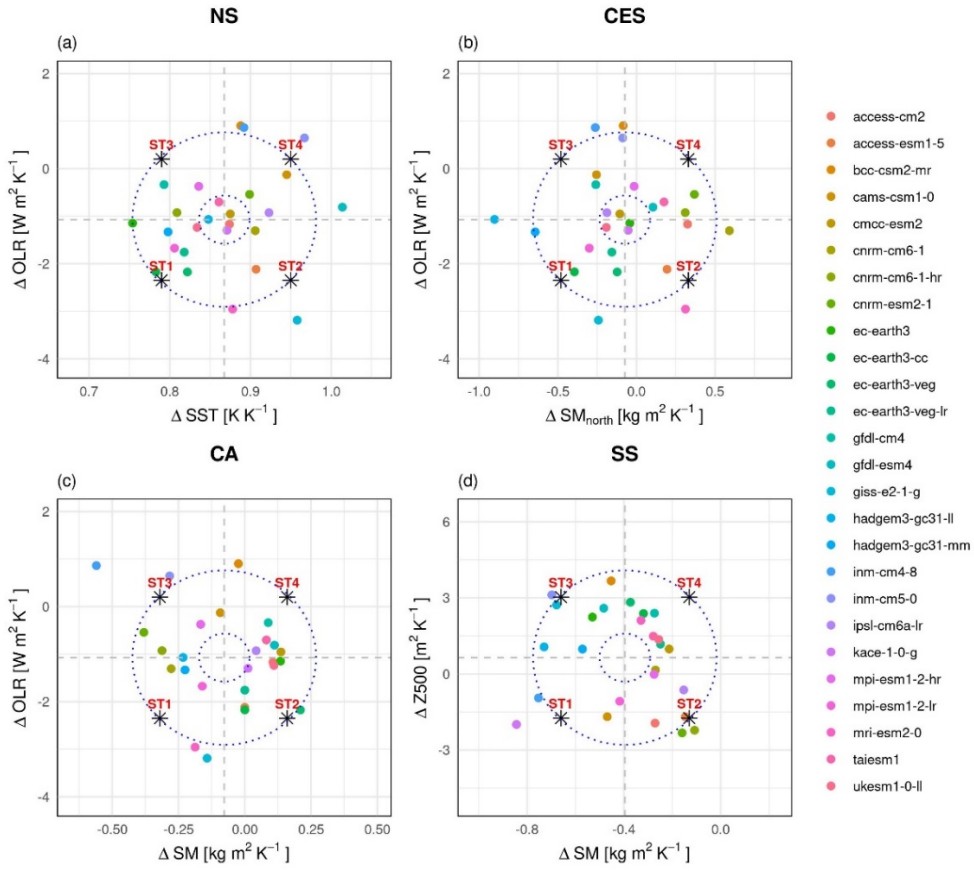





**Figure 4: Drivers' responses scaled by global warming (GW) for each GCM (colour circles) across SSA regions: (a) NS, (b) CES, (c) CA, and (d) SS. Black stars indicate the four storylines of ΔTXx derived from extreme responses of the two most influential drivers (see Eq. 2). Dashed black ellipses indicate the 80% confidence region, obtained by fitting a bivariate normal distribution to the GCM responses. Each quadrant displays the combination of the two drivers associated with each storyline.**

For instance, ST1 for the NS region (Fig. 4a) is characterised by lower-than-MMM changes in both N3.4 and SACZ, while ST4 represents the opposite pattern. Similarly, ST2 and ST3 correspond to opposing changes in these two drivers. The specific combination of drivers for each ST and region, as obtained from Fig. 4, is summarised in Table 2.

| Region | ST1 | ST2 | ST3 | ST4 |
|---|---|---|---|---|
| NS | Low ΔN3.4 + Low ΔSACZ | High ΔN3.4+ Low ΔSACZ | Low ΔN3.4+ High ΔSACZ | High ΔN3.4+ High ΔSACZ |
| CES | Low ΔSM$_{north}$ + Low ΔSACZ | High ΔSM$_{north}$ + Low ΔSACZ | Low ΔSM$_{north}$ + High ΔSACZ | High ΔSM$_{north}$ + High ΔSACZ |
| CA | Low ΔSM$_{CA}$ + Low ΔSACZ | High ΔSM$_{CA}$ + Low ΔSACZ | Low ΔSM$_{CA}$ + High ΔSACZ | High ΔSM$_{CA}$ + High ΔSACZ |
| SS | Low ΔSM$_{SS}$ + Low ΔZ500 | High ΔSM$_{SS}$ + Low ΔZ500 | Low ΔSM$_{SS}$ + High ΔZ500 | High ΔSM$_{SS}$ + High ΔZ500 |

**Table 2: Combination of drivers selected to create the corresponding storylines (ST1 to ST4) of ΔTXx for each region.**

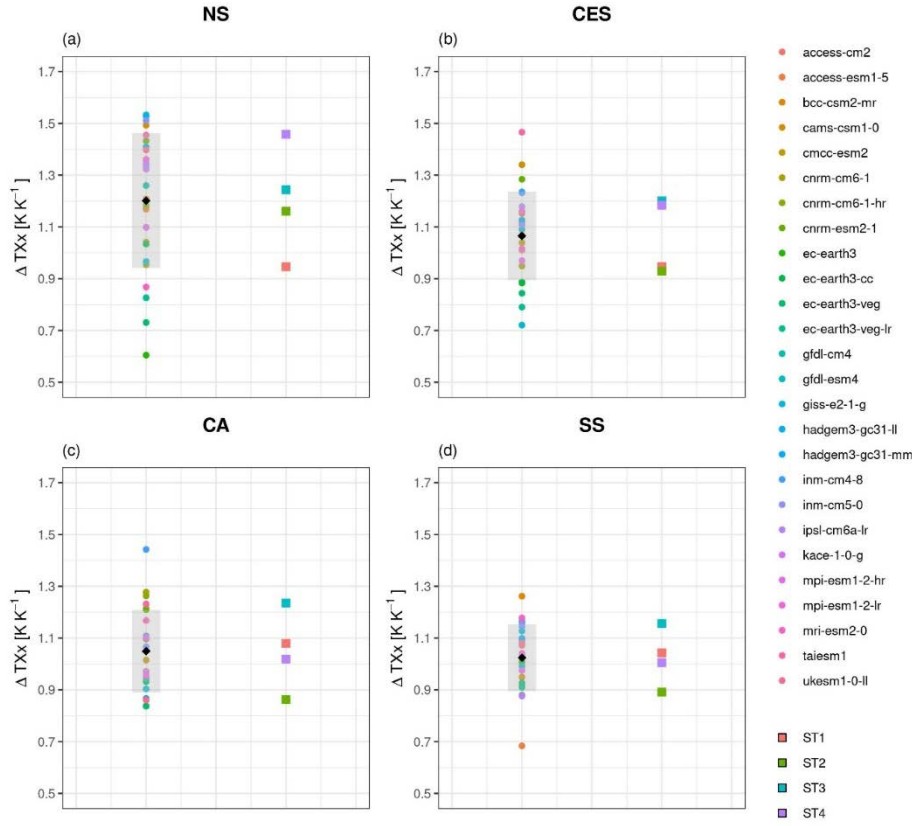



**Figure 5: Summer (DJF) TXx responses (2070–2099 minus 1979–2014) per degree of GW (K/K) for each GCM (coloured circles) and for each SSA region: (a) NS, (b) CES, (c) CA, and (d) SS. The black diamond denotes the MMM and the grey-shaded area indicates its one standard deviation. The four storylines are depicted with filled colour squares.**

Figure 5 illustrates the scaled summer $\Delta$TXx for each GCM (coloured circles), including the MMM (black diamond) and its one-standard-deviation range (grey shading), as well as the reconstructed storylines of $\Delta$TXx (coloured squares) based on the combination of drivers' responses. This figure also reveals which combination of drivers leads to the largest and smallest warming in each region, i.e. the worst-case and best-case scenarios, respectively. Overall, TXx warming appears unavoidable, as even the best-case scenario shows an increase of ~0.9 K/K across all regions. Moreover, the TXx warming response in the worst-case storyline is 29% to 54% higher than in the best-case scenario. The inter-storyline variability reasonably encompasses the range of uncertainties in $\Delta$TXx projections, represented by the grey-shaded areas in Figures 5a–d. In the remainder, the two remaining storylines will not be discussed, as they exhibit an intermediate result.

In NS, the largest warming in TXx (ST4, purple square; Fig. 5a) results from the combination of a warming in the tropical Pacific and a strengthening of the SACZ relative to the MMM changes. This storyline determines a 54% greater increase in $\Delta$TXx compared to the opposite combination of drivers' responses. CES and CA storylines are constructed using similar drivers (as seen in Table 1). However, the response in $\Delta$TXx differs between the two regions. For CES, $\Delta$SACZ is the only driver with significant influence on $\Delta$TXx (Table 2; Fig. 5b). This is reflected in the separation between the storylines. ST1 and ST2 are associated with a SACZ weakening, whereas ST3 and ST4 correspond to weak SACZ intensification, with the latter yielding an additional ~29% increase in $\Delta$TXx. Comparatively, the difference in $\Delta$TXx between ST1 and ST2, as well as between ST3 and ST4, is negligible. This pattern highlights that the spread of $\Delta$TXx projections over CES is primarily driven by SACZ variations, while $\Delta$SM$_{north}$ does not play a significant role. In contrast, in CA, the combination of strong drying and an intensification of the SACZ leads to a $\Delta$TXx warming ~44% higher than that associated with the opposite storyline (Fig. 5c). Finally, the storyline characterised by the largest warming in SS $\Delta$TXx (ST3, blue square; Fig. 5d) is determined by the combination of enhanced drying and anticyclonic activity relative to the MMM, with an additional 30% increase in $\Delta$TXx warming compared to the best-case storyline (ST2, green square).

For better interpretation of the differences in the storylines of $\Delta$TXx, Figure 6 shows the composite difference of the spatial patterns of $\Delta$TXx (shading) and $\Delta$Z500 (contours) between the GCMs following the worst- and best-case storyline of regional $\Delta$TXx. Enhanced warming under the worst-case outcome is evident across all regions, particularly in NS. In contrast, in CES, the $\Delta$TXx differences between extremal storylines are small, consistent with the poorer MLR performance and the weak influence of one of its drivers. The worst-case scenarios of each region are also accompanied by distinctive circulation anomalies, featuring Rossby wave trains with different pathways and latitudes depending on the region, which are consistent with the enhanced regional $\Delta$TXx responses. Likewise, in NS, CES, and to a lesser extent CA, the drivers associated with the largest warming in $\Delta$TXx lead to anticyclonic anomalies over the South Atlantic Ocean. This is consistent with Suli et al. (2023), who found that HWs in these regions are triggered by shifts/intensification of the subtropical semi-permanent high-




pressure systems. The influence of the subtropical anticyclone is missing in the worst-case storyline of SS, where HWs are
related to co-located anticyclonic anomalies (blocking) and jet meandering.

Overall, these findings underscore the importance of identifying region-specific drivers and exploring physically plausible
scenarios beyond the MMM. For all regions, we find that changes in both thermodynamic (SM, N3.4) and dynamic (Z500,
convection-related SACZ) drivers contribute to the spread of future projections in regional ΔTXx, stressing the importance of
understanding dynamical aspects of climate change in the region.

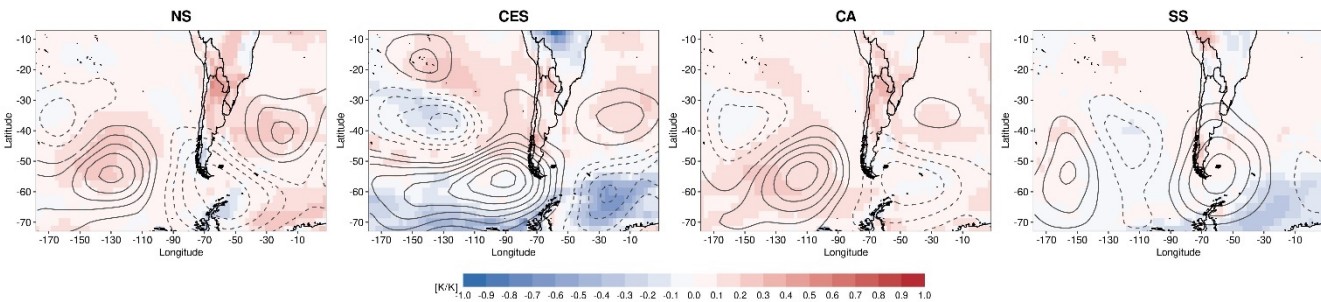

**Figure 6: Differences between the composites of projected changes in TXx (shading, K/K) and Z500 (in contours, m/K) for the GCMs
with the strongest (worst-case storyline) and weakest (best-case storyline) warming in regional TXx: (a) NS, (b) CES, (c) CA, and
(d) SS. Contours are shown every 1 m/K, with solid (dashed) black lines representing positive (negative) ΔZ500. Values are expressed
per degree of GW.**

## 4 Conclusions and discussion

In this study we assessed the sources of uncertainty in maximum summer temperature (TXx) projections over Southern South
America (SSA) using historical and future simulations of 26 global climate models (GCMs) from the Coupled Intercomparison
Project Phase 6 (CMIP6). To do so, we applied for the first time the storyline approach to TXx changes (ΔTXx) in four
subregions of SSA: northern SSA (NS), central-eastern SSA (CES), central Argentina (CA) and southern SSA (SS). Storylines
were created for each region based on the climate change responses in key drivers of ΔTXx, including mid-tropospheric
circulation (Z500*, ridging activity), regional soil moisture (SM), sea surface temperature in the Niño 3.4 region (N3.4), and
variations in the South Atlantic Convergence Zone (SACZ). The main results can be summarised as follows:

- Future changes in the drivers of SSA temperature extremes: The multi model mean (MMM) changes at the end of the
  century (2070–2099) reveal a strong soil drying in central northern SA and along the Andes mountains, whereas the
response in SM content over central SSA remained highly uncertain. The central-eastern Pacific is projected to warm by
  up to 4°C above historical values, resulting in enhanced convection over the region. Moreover, changes in Z500* feature
  a Rossby wave pattern with enhanced anticyclonic conditions at high latitudes in southern SA. However, these elements
  can have competing effects in regional changes in ΔTXx, and their responses to climate change are affected by substantial
  uncertainty, which propagates to that in the future projections of regional ΔTXx.





• Relevant regional drivers of SSA temperature extremes: Several physically coherent drivers were considered, and the most relevant combinations were identified for each SSA subregion using a multi-linear regression (MLR) framework. The results indicate that different drivers influence ΔTXx depending on the region. In NS, ΔTXx was primarily linked to remote influences (ΔN3.4 and ΔSACZ). For CES and CA, both remote and local factors contributed, namely ΔSM and ΔSACZ. In contrast, in SS, the projected warming was mainly explained by proximate processes, particularly regional

soil drying and changes in mid-tropospheric circulation. The MLR accounted for 35% to 56% of the variance in regional ΔTXx, with both predictors showing significant contributions, except for SM in CES, where its influence was negligible. Given the presence of multiple relevant drivers in most regions, a storyline framework was used to explore the combined effects of their projected changes.

• Storylines of changes in regional temperature extremes: The magnitude of the projected summer warming in regional ΔTXx depends on specific combinations of its climate drivers, which vary from region to region. The storylines in ΔTXx effectively represent the inter-model variability of future changes in TXx and help explain the physical mechanisms behind their uncertainties. Differences in ΔTXx between the best- and worst-case storyline ranged from 29% to 54%, with NS region showing the greatest sensitivity to drivers' combinations. In this region, the highest warming in ΔTXx resulted

from enhanced central-eastern Pacific warming with respect to the MMM, which is associated with El Niño events, and SACZ intensification, leading to a ~0.5 K/K (over 50%) increase compared to the opposite combination of drivers' responses. In SS, the strongest warming in ΔTXx was linked to enhanced soil drying and anticyclonic activity, while in CA intensified soil drying and SACZ intensification resulted in the worst-case storyline. Finally, in CES, the warming in ΔTXx is primarily driven by the strengthening of the SACZ, with soil moisture playing a negligible role.

Therefore, future projections of TXx in SSA show spatial variations, and their uncertainties are governed by different drivers (depending on the region), which often reflect a combination of thermodynamic and dynamical aspects of climate change. As the drivers of uncertainty in TXx projections vary across regions, it is also natural to ask if they also depend on the specific aspect of the extreme event that is being scrutinized (i.e. the extreme index). The results reveal that the drivers of regional ΔTXx show varying skill to explain uncertainties in future projections of more complex metrics, such as the percentage of

summer days exceeding the 90th percentile (TX90). The regional responses of TX90 to the aforementioned drivers are generally weaker than those in TXx, with most regions showing non-significant signals (see Table S2). This is also accompanied by a reduction in the explained variance ranging from 0.13–0.35 over most regions (not shown). Statistically significant responses were only found in CA and SS, where enhanced ΔTX90 was associated with strong regional soil drying and intensified anticyclonic activity, respectively. However, the remaining drivers of ΔTXx in these regions did not show a

significant response in TX90, and none of the ΔTXx drivers in NS and CES explained a significant fraction of ΔTX90 variance. Similar results are found for the HW attributes (i.e. HW duration, areal extent, and intensity; Table S2) derived from a spatio-



temporal tracking HW algorithm (Sánchez-Benítez et al. (2020) applied to characterise HWs in Argentina (Collazo et al., 2024).

These differences suggest that the drivers of maximum absolute summer temperature differ from those based on relative thresholds like TX90, arguably reflecting different sensitivities to changes in the mean and variability of extremes (Barriopedro et al., 2023, and references therein). Garrido-Perez et al. (2024) found similar results when analysing extreme temperature responses in the Iberian Peninsula from a variety of indices. They suggested that the complexity of the assessed variable may lead to weaker responses to the drivers. Regardless of the causes, the observed differences indicate that the drivers and associated storylines of extremes should not be generalised to all indices and attributes. The use of emerging tools, including

artificial intelligence (e.g., Pérez-Aracil et al., 2024) may help uncover additional drivers on extended spatio-temporal scales, and the differences across extreme indices.

**Data availability**

ERA5 reanalysis data is freely available at the Copernicus Climate Change Service Climate Data Store: https://cds.climate.copernicus.eu/datasets?q=reanalysis&limit=30.

The Coupled Model Intercomparison Project Phase 6 (CMIP6) data for this study have been obtained from the ESGF website: https://esgf-metagrid.cloud.dkrz.de/search/cmip6-dkrz/

**Author contribution**

DB and RGH conceived of the presented idea. SS conducted the experiments and prepared the figures. SS, DB, RGH and SC contributed to the interpretation of the results. SS led the writing of the original draft with contributions from DB and RGH.

DB, RGH, SC, AS, and MR provided critical feedback, helping with the organisation, revision, and editing of the manuscript until its final version.

**Competing interest**

The authors have declared no competing interests.

**Acknowledgments**

This research was supported by predoctoral grant program from the Comunidad de Madrid (No. PIPF-2022/ ECO-25310), PIP0333 and 20020220200111BA projects from the National Scientific and Technical Research Council (CONICET) and University of Buenos Aires (UBA). This work was also supported by the European Union's Horizon 2020 research and



innovation program under the Marie Sklodowska-Curie grant agreement No 847635 (UNA4CAREER) through the SAFETE project (code 4230420). The authors are also grateful to project CLINT (Grant Agreement No. 101003876).

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
