# Peer review of "Storylines of extreme summer temperatures in southern South America"

_EGUsphere, 2025_

## Author Comment (AC1)

**Response to reviewers of Suli S. et al.: Storylines of extreme summer temperatures in southern South America.**

We are grateful to the referees for the time they have taken to review the manuscript. We appreciate their comments, which have helped improve the manuscript. Our replies (normal font) are given after their comments (bold font). We have updated the manuscript following most of their recommendations.

**Referee #1**

**Line 63: Where is Iberia located?**

Iberia is located in southwestern Europe and comprises Spain and Portugal. To clarify it, we have changed this to *"Iberian Peninsula"* (line 64).

**Lines 114-115: How did you interpolate the data? Have you considered interpolating all data to the coarser resolution? It should be explained as well.**

We interpolated all datasets to the coarser resolution of the GCMs, which corresponds to $2.5° × 2.5°$, as shown in Table S1. In the text, this is explained as: *"A common 2.5° x 2.5° horizontal grid and the austral summer season (…)"* (line 118).

In the original manuscript (lines 119–121 of the revised manuscript) we explained that we used a bilinear interpolation. In the revised version we have included the following information: *"Bilinear interpolation was used for TX, SST, Z500 and OLR data, while a conservative remapping was applied to SM data to avoid spurious values (Jones, 1999)".* Conservative remapping assigns values based on the overlapping area between the original and new grids, ensuring that the overall field is conserved during interpolation.

**I recommend not abbreviating southern South America (SSA) as it may be confused with southern southern South America (SS).**

We understand the reviewer's concern regarding the potential confusion between southern South America (SSA) and southern subregion (SS). However, for consistency with our previous work (Suli et al., 2023) and to avoid excessive repetition of "southern South America" throughout the manuscript (this region is cited more than 35 times), we prefer to retain the acronym SSA to denote the broader region.

**Section 2.3: Please specify which drivers are local and which are remote (and why, preferably) in this Section rather than in the Introduction.**

We have clarified in Section 2.3, adding between brackets the driver type: "local driver" and "remote driver" (e.g., *"Sea surface temperature in Niño 3.4 region (N3.4, remote driver)"*). We also added the following lines in the same section (lines 153-154 of the revised manuscript): *"Local drivers are proximate factors that directly influence regional temperature, whereas remote drivers represent large-scale influences or teleconnections affecting the region".*

**The manuscript has too many abbreviations. I recommend adding a table with all of them to support the reader.**

We have reduced the number of abbreviations (e.g., Pc90, AR6, HW) where possible to improve readability. However, we believe that including a full table of acronyms is not standard procedure and would not necessarily make the manuscript easier to follow.

**I suggest using grey dots in areas where changes are statistically significant in Figure 2.**

Thank you for the suggestion. We tested plotting grey dots in statistically significant regions (see Figure R1). For some variables (e.g., ΔTXx), the entire domain is statistically significant, and for others (e.g., ΔZ500*) highlighting significant regions would hide the main circulation patterns. Therefore, for the sake of clarity, we prefer to keep the original plot.

[Figure]

*Figure R1.* Multi-model mean (MMM) summer (DJF) changes in (a) Maximum Temperature (TXx, K), (b) Soil Moisture (SM, kg m⁻²), (c) Sea Surface Temperature (SST, K), (d) Outgoing Longwave Radiation (OLR, W m⁻²) and (e) Geopotential Height at 500 hPa with the zonal mean removed (Z500*, m). Changes are computed as the difference between the periods 2070–2099 and 1979–2014. Grey dots indicate areas where changes are statistically significant at the 95% confidence level, based on a two-tailed t-test.

**Lines 276-278. This sentence should be tied to the last paragraph.**

We have revised the text to improve the flow of the paragraph, better connecting the storylines with the regional ΔTXx examples in Figure 4 and the subsequent description of Figure 5. In the revised manuscript, the paragraph now reads as follows (lines 303–304): *"Each storyline characterises the summer ΔTXx as the combined response in $b_x$ and $c_x$ (see Eq. 2), yielding distinct patterns of warming depending on how the two drivers change."*

**I am not sure if Table 2 is relevant to the manuscript. Results are not discussed in**

**the text.**

We understand that Table 2 is not extensively discussed in the text. However, we believe it provides a useful summary of the storyline results. Given the number of regions and the four driver combinations per region, the table allows readers to quickly identify the best and worst combinations of drivers for each storyline, which could be confusing if described only in the main text.

**Line 297: What does weak ZCAS intensification mean?**

According to a comment from Reviewer #2, we revised the interpretation of this index and now refer to it as OLRg. The definition based on the OLR gradient remains unchanged, but it is now interpreted in a broader sense, involving changes in atmospheric stability. In this context, a "weak OLRg intensification" refers to a small strengthening of the OLR gradient (as observed in ST3 and ST4 of CES region; see black stars in Figure 4). This positive $\Delta$OLRg indicates a slight increase in OLR over the equatorward box and/or a decrease over the poleward box of Fig. 3b, d, f, thereby inhibiting convection in the former and reinforcing it in the latter. Such a pattern could result from an intensified or extended subtropical Atlantic anticyclone, or from changes in convection associated with the SACZ.

**The conclusions are quite long, and the writing is confusing. I recommend shortening it to something between 700 and 900 words.**

Despite the new analyses performed in the revised version, we have slightly shortened Section 4 (Conclusions and discussion), as requested, while keeping its overall structure. The revised version keeps the main results of the study: (i) projected changes in the key drivers, (ii) the identification of the most relevant drivers explaining the sources of uncertainty in $\Delta$TXx, and (iii) the storylines of $\Delta$TXx. This is followed by discussion comparing our findings with more complex extreme temperature indices.

---

## Author Comment (AC2)

**Response to reviewers of Suli S. et al.: Storylines of extreme summer temperatures in southern South America.**

We are grateful to the referees for the time they have taken to review the manuscript. We appreciate their comments, which have helped improve the manuscript. Our replies (normal font) are given after their comments (bold font). We have updated the manuscript following most of their recommendations.

**Referee #2**

**The authors provide a storyline analysis of summer absolute maximum temperature (TXx) over four regions of southern South America: northern, central-eastern, central Argentina, and southern areas. The storyline approach allows them to explain 56% of inter-model spread in projections of TXx. I found the analysis very interesting and highly relevant to better understand future projections of climate in South America. However, I find that the authors could improve the robustness of their results if they considered the uncertainty in the remote drivers due to internal variability. Methodologies to account for this uncertainty in storyline studies have been presented (e.g. in an article cited here, Mindlin et al. (2020), based on which other analysis have estimated uncertainty in the responses). I suggest major revisions and that the authors consider the suggestions to ensure that the statistical methods of storyline analysis are performed as robustly as possible. It is fundamental to ensure that the regression captures differences across models that are representative of climatological changes and not artifacts of model biases or internal variability.**

We would like to thank the reviewer for their thoughtful comments. He/She raised important points that will help readers better understand and evaluate our study. We have carefully considered the issues mentioned by the reviewer, paying special attention to those related to internal variability and model biases. We provide detailed responses and additional clarifications in the comments below.

**The authors put their analysis in context and cite recently published literature illustrating there is uncertainty in extreme temperatures in the region of study (IPCC, 2023), justifying this analysis. In lines 55-70 previous storyline analysis are referenced. However, the authors do not reference Mindlin et al. (2024) who analysed climate impact drivers in the same region (https://doi.org/10.1016/j.cliser.2024.100480). It is particularly interesting that in that paper, the authors show that with a "low Pacific warming" storyline (which would correspond to "Low ΔN3.4 +") the maximum temperature in the extended warm season ONDJFM is high more extreme than under a high Pacific storyline. This is the case both in the north of SS, CCH and the west of CES. It would be interesting if the authors could put their results in the context of this other storyline analysis of temperature indicators for the same region. The authors mention Garrido-Perez et al. (2024) on Iberia, but I don't understand how this has to do with their study.**

Thank you for the reference. We have now included Mindlin et al. (2024) in the introduction as a recent study applying the storyline framework over southern South America (SSA, lines 64–65): *"Similarly, Mindlin et al. (2024) applied the storyline approach to examine climate impact drivers over southwestern SA, including temperature-based indices"*. We kept the reference to Garrido-Pérez et al. (2024) as it provides another recent example of the storyline methodology applied to temperature-related projections. Although it focuses on a different region (Iberian Peninsula), we found similarities between the two studies in the distinct behaviour of temperature indices.

To deep further into the N3.4 question, we first computed the storylines of TXx using the ΔN3.4 and ΔOLRg drivers (for details on the OLRg index, see the comment related to the SACZ index). Figure R2 displays only ST1 (Low ΔN3.4 + Low ΔOLRg) and ST2 (High ΔN3.4 + Low ΔOLRg) to isolate the TXx response to Pacific SST changes while keeping OLR gradient variations low. These results are partly consistent with those reported by Mindlin et al. (2024): Under low ΔN3.4 conditions, a slightly warmer response is found north of 35°S (northern grey box in Fig. R2). This region includes much of the CCH region, which was not analysed due to model biases. The signal weakens farther south (southern grey box in Fig. R2, which captures only the northern SS region). Conversely, in the northern SSA region (NS; purple box in Fig. R2), we found the opposite response, and a more pronounced sensitivity to Pacific SST warming. Therefore, there is not a full correspondence between the regions analysed here and those of Mindlin et al. (2024).

[Figure]

*Figure R2*. Absolute maximum temperature responses (ΔTXx) scaled by global warming (GW, K/K) under two storylines (STs): ST1 (Low ΔN3.4, Low ΔOLRg) and ST2 (High ΔN3.4, Low ΔOLRg). STs were constructed as the difference between the future (2070–2099) and historical (1979–2014) periods during the austral summer (DJF). Grey boxes indicate the approximate regions analysed in Mindlin et al. (2024), while purple boxes show the Northern region (NS) of SSA used in this study.

Secondly, we analysed TXx responses for DJF using the ΔCP and ΔEP drivers defined in Mindlin et al. (2024) (Fig. R3). The general spatial patterns are comparable to those described above, but the contrasting response between low and high Pacific warming reported by Mindlin et al. (2024) is not clearly evident in this analysis. Indeed, the variance explained ($R^2$) by ΔCP and ΔEP drivers in our SSA regions is 0.2–0.5 lower than the $R^2$ obtained with our selected drivers (Table 1 of the manuscript). The differences likely reflect the sensitivity of the regression-based storylines to the selected targets (TX

vs TXx), seasonal definitions (extended summer, ONDJFMA vs high summer, DJF), and regional domains (Fig. R3 vs Fig. 1 of the main text). This has been mentioned in Section 3.3 (lines 327–330): *"Differently, Mindlin et al. (2024) found a larger increase in October-April mean TX over southwestern SA under a low Pacific warming storyline (i.e. a relative cooling of both eastern and central El Niño), whereas a high Pacific warming produced the opposite response. The discrepancies with our findings may stem from differences in the selected drivers, target variables, seasonal definitions, and/or regional domains"*.

[Figure]

*Figure R3*. Absolute maximum temperature responses (ΔTXx) scaled by global warming (GW) [K/K] under low and high Pacific warming storylines (Low CP + Low EP and High CP + High EP, respectively). CP and EP denote Central Pacific and Eastern Pacific warming, respectively, following the definitions in Mindlin et al. (2023). The storylines (STs) were constructed using the difference between the future period (2070–2099) and the historical period (1979–2014), considering the summer season (DJF). Grey boxes indicate the approximate regions analysed in Mindlin et al. (2024).

**105 Why did the authors use one ensemble member per model? It would be useful to understand the uncertainty in the local drivers. In particular, it would be good to know that the soil moisture response is robust and that the uncertainty is not associated with internal variability.**

We carefully considered the possibility of including multiple ensemble members per model. Our analysis combines several variables (TX, SM, OLR, SST) with different temporal resolutions (daily and monthly) and covers both historical and SSP5-8.5 simulations. Unfortunately, only a limited subset of models provides all these datasets and scenarios for multiple realisations. In other words, the number of models that meet all these requirements decreases substantially when more than one member per model is

demanded (see Fig. R4). For example, if we restricted the analysis to models with at least three ensemble members, which would allow some filtering of internal variability, the available sample would be reduced to only 38% of the original set (10 out of 26 models), affecting the robustness of the MLR model. Therefore, we retained one ensemble member per GCM in order to maximise the size and diversity of the multi-model ensemble, which we considered essential for a more robust storyline assessment. A larger ensemble allows a better representation of the full spread of responses, although this approach does not remove internal variability. To clarify this issue in the manuscript, we have added the following lines in the revised version (lines 110–113): *"To maximise the number of models, we considered one ensemble member per GCM. Although this strategy does not remove internal variability—an issue that may require large ensembles (e.g. Deser et al. 2020 and references therein)—it does increase the sample size available for constructing the storylines (requiring three or more ensemble members per model would have reduced the ensemble to less than half its original size)"*

[Figure]

*Figure R4.* Distribution of the number of members per model. The last bar ('Total') represents the overall number of models analysed in this study.

To further evaluate the robustness of the soil moisture (SM) response, we performed an additional analysis using all available ensemble members for each model (Fig. R5). In this case, we computed $\Delta$SM for each simulation to assess how internal variability contributes to the spread of multi-model responses. For models with a large ensemble (>10 members), we limited the analysis to the first ten members to maintain balance across models. The results show that the spread within individual models (intra-model variability) is at least three times smaller (see also Table R1) than the spread across models (inter-model variability), suggesting that the uncertainty in SM projections is primarily driven by model spread rather than by internal variability (Fig. R5). We obtain similar results if the inter-model spread is computed using only the models that have more than one ensemble member. Similar results are also obtained when analysing the $\Delta$SM$_{north}$ driver (covering both NS and CES regions). These results confirm that the spread is

largely dominated by the forced signal, supporting the robustness of the SM changes inferred from a single member.

[Figure]

*Figure R5.* Distribution of soil moisture (SM) changes scaled by global warming (GW) (ΔSM; kg m⁻² K⁻¹) for GCMs across SSA regions (from top to bottom panel: northern SSA (NS), central-eastern SSA (CES), central Argentina and northern Argentinian Patagonia (CA), central Chile (CCH), southern SSA (SS)), considering all available members. Models with more than one ensemble member are shown in bold and for those with more than ten members, only the first ten were considered.

| Region | Internal variability (11 models) | Inter-model spread (26 models) |
|---|---|---|
| NS | 0.11 | 0.38 |
| CES (box$_{north}$) | 0.10 | 0.33 |
| CES | 0.09 | 0.29 |
| CA | 0.05 | 0.19 |
| SS | 0.05 | 0.21 |

*Table R1.* Internal variability and inter-model spread of projected ΔSM for each SSA region. Internal variability is computed as the mean intramodel deviation of ΔSM responses for models with more than one ensemble member.

**115-120 The authors use absolute values of maximum temperature as an index. What I understand from other storyline analysis based on precipitation changes, is**

**that climatological changes are analysed and hence the regression analysis captures the difference across models that can be explained due to uncertain changes in each driver. In this case, the absolute value of maximum temperature in the models might differ due to model biases and hence the biases can be confounding the signal. Could you please explain how in this analysis the potential biases in models are addressed?**

We understand that the use of the term "absolute summer maximum of TX (TXx)" may have caused some confusion. This is a common extreme index employed in the literature, herein defined as the warmest TX value of the summer season. For the storyline analysis (lines 173–175 in Sect. 2.4), we focus on changes in TXx ($\Delta$TXx), computed as the difference between the climatological TXx means for the far future (2070–2099) and the historical period (1979–2014). By definition, the storyline (i.e. the predictand in the multi-linear regression) is constructed for $\Delta$TXx in order to capture the inter-model spread in the projected climate change signal, rather than for the absolute climatological values TXx. Bias correction is not applied, as we assume that model biases remain constant in the future (i.e. they are independent of the climate scenario). Under this assumption, $\Delta$TXx would be largely unaffected by biases, although TXx itself would be. To further address this, we have analysed the correlation between $\Delta$TXx (scaled by GW) and TXx biases for each SSA region (Fig. R6). No statistically significant relationships were found ($p < 0.05$), except for CCH, which had already been excluded from our study. This indicates that model biases in the mean cannot explain future summer temperature changes, and hence they would have little influence in the multi-model spread of mean projections. To clarify how potential biases are addressed in the study, we have added the following lines in the revised manuscript (lines 200–201), supported by Figure R6 (new Figure S3 in the supplement): *"In the construction of storylines, we assume that model biases remain constant in the future, and therefore do not substantially influence the climate change signals. Figure S3 supports this hypothesis by revealing no statistically significant relationship between model biases in TXx and their projected changes, $\Delta$TXx."*

[Figure]

*Figure R6.* Summer TXx changes ($\Delta$TXx) scaled by global warming (K/K) against TXx biases [K] for each GCM (colour circles) and SSA region (panels). The Pearson correlation coefficient is shown in the upper-right corner, with an asterisk denoting statistically significant correlations at the 95% confidence level.

**125 Regionalization is based on present climate. However, the storyline analysis allows the authors to better understand the uncertainty in the forced response. The signal of change could not overlap with the climatological regions. Could you please better justify the regionalization being based on the reference climatology and not on the regions where the projected change is most uncertain?**

The regionalisation employed in this study is based on the spatial distribution of heatwave days during the historical period (1979 – 2023). This choice was primarily chosen for continuity with our previous work (Suli et al., 2023), in which the physical mechanisms triggering heatwaves over the resulting southern South America (SSA) regions were characterised. Maintaining this definition enables a direct comparison between past and present analyses, preserves the spatial coherence in regional heatwave occurrence, and ensures consistency in the selection and interpretation of the drivers. If the regions were instead defined according to where the projected changes are more uncertain -a legitimate choice-, the signal of change would be likely stronger. However, this would emphasise uncertainty (likely coming from different sources) over understanding of regional phenomena and their underlying drivers. Our intention in this study is to keep a physically consistent framework that clarifies how different drivers contribute to regional TXx changes. This has been stated in lines 145-148 of the revised manuscript: *"Note that this regionalisation aims to provide a robust characterisation of regional extremes rather than to identify areas of homogeneous changes or high uncertainty in future projections. The latter approach would maximise the ensemble spread at regional level but would also shift the focus away from the behaviour of spatially coherent regional phenomena and their underlying drivers"*.

**145 – 150 On the selection of remote and local drivers, the authors chose to characterize SACZ changes in terms of SACZ intensity. Zilli et al. (2019) and Zilli and Carvalho (2021) reported a poleward shift of the SACZ and methodologies of convergence zone shift have been proposed and used in other storyline analysis. In this analysis of storylines of for the South Pacific Convergence Zone the authors illustrate that the poleward shift and strength of such poleward shift explains a great part of the model uncertainty https://doi.org/10.1175/JCLI-D-21-0433.1 I am concerned that by defining and index based on the difference of OLR between the two proposed boxes as a way of capturing the intensity of the SACZ the authors in this study miss the most important component of the uncertainty in the SACZ chanages and its associated impacts. This would be captured by an index that captures spread in the poleward shift of the SACZ. Indeed in their results, only for CES the sensitivity pattern is significant in a region expected to be affected by the SACZ. Could you please illustrate what is the model spread of the SACZ change that this index captures?**

Thank you for the references. The OLR-based index was not originally designed to strictly represent the SACZ. Our initial goal was to identify meaningful drivers of regional changes in TXx through linear regressions, which revealed a dipole in OLR. In a subsequent analysis, we confirmed that this OLR gradient is also related to the SACZ, and we therefore referred to it as the SACZ driver, although we acknowledge that this is not the standard SACZ definition, and hence this driver has been renamed in the revised

version. Figure R7 illustrates the correlation of this driver (the OLR gradient, OLRg) with precipitation (first panel) and Z1000 (second panel) for the multi-model ensemble over the historical period. The OLRg index reflects a southward shift in SACZ-related precipitation as well as variations in the extension of the subtropical Atlantic anticyclone, both of which would influence regional convection and subsidence patterns. These results suggest that OLRg would be affected by changes in atmospheric stability caused by either the subtropical Atlantic anticyclone or the SACZ.

[Figure]

*Figure R7.* MMM Pearson correlation coefficients of the summer (DJF) series of the OLRg driver and 2D fields of: left) precipitation; right) Z1000 during the historical period (1979–2014). Boxes indicate the regions used to construct the OLR gradient and stippling areas denote regions where at least 66% of the models show significant correlations (p < 0.1) of the same sign.

Consequently, to address the reviewer's concern, we have renamed the OLR-based index (formerly SACZ; now simply OLRg) and refined its interpretation. While we retain the original definition (the OLR difference between the two boxes), we now clarify that this index should be interpreted as an indicator of changes in atmospheric stability and large-scale circulation over the subtropical South Atlantic. This framing avoids over-interpreting OLRg as a direct proxy for the SACZ intensity and provides a more accurate description. The revised manuscript includes a new Figure (Figure S2) in the supplement and is clarify as follows (lines 168-171): *"This OLR gradient reflects regional convection patterns linked to variations in atmospheric stability. Additional analyses (Figure S2) confirm that, on interannual scales, a strengthening of the OLR gradient is associated with an intensified or zonally elongated subtropical Atlantic anticyclone, as well as with poleward shifts in SACZ-related precipitation (Liebmann et al., 2004)."*

We note that our index is still sensitive to meridional displacements of the SACZ (a poleward shift would decrease OLR in the southern box and increase it in the northern one, thus increasing the gradient). However, we agree that meridional shifts of the SACZ would be better diagnosed by using indices designed to track convergence areas. Moreover, the multi-model mean response in precipitation (or OLR) and its inter-model spread is not dominated by a simple meridional pattern, at least in CMIP6. The

uncertainty pattern is more complex than that, which agrees with the uncertain response of the SACZ itself.

To accommodate the above changes, we have also revised Section 3.2. We now begin by presenting the TXx sensitivity patterns (Figure 3 in the revised manuscript) in order to select meaningful remote and local drivers with large regional TXx responses. Regarding the reviewer's question on model spread, a new Table S2 has been added to the supplement (see also Table R2 below). It compares the inter-model spread of future changes in the selected drivers with an estimate of internal variability (based on the standard deviation of the detrended annual series), following Mindlin et al. (2020). The ratio of variances is assessed with an F-test. For all drivers, the uncertainty in the responses across the multi-model ensemble is clearly distinguishable from (significantly larger than) the internal variability ($p<0.1$). For the specific case of the OLRg driver mentioned by the reviewer, the spread is substantial implying changes of opposite sign, and consistent with the uncertainty in convection-related changes over that region. Notably, 70% of the models exhibit statistically significant changes in the OLRg index.

| Driver | Internal variability | Inter-model spread | Inter-model spread / Internal variability |
|---|---|---|---|
| N3.4 | 0.07 | 0.45 | 6.42[*] |
| OLRg | 3.13 | 13.55 | 4.32[*] |
| Z500$_{HL}$[*] | 27.98 | 51.71 | 1.85[*] |
| SM$_{SS}$ | 0.06 | 0.78 | 13.00[*] |
| SM$_{CA}$ | 0.09 | 0.54 | 6.00[*] |
| SM$_{north}$ | 0.18 | 2.20 | 12.22[*] |

*Table R2.* Internal variability (MMM standard deviation of the detrended series) against inter-model spread (standard deviation of the projected changes) for the selected drivers of each SSA region. All values are scaled by global warming. The last column represents the ratio of uncertainty to internal variability, with an asterisk indicating significant differences at $p<0.1$ after an F-test.

**Regional soil moisture is a very hard variable to work with. Did the researchers explore the performance of the models in simulating the climatology and variability of soil moisture? A storyline analysis based on this variable can be hard to trust if the variable itself is not well represented. Is the spread in soil moisture change distinguishable between models (i.e. what is the spread in climatological soil moisture in the models? Is the spread significant with respect to internal variability?). The significance of remote driver changes was not evaluated in Zappa and Shepherd (2017) but has been reported in storyline analysis like Mindlin et al. (2020), appendix.**

We acknowledge that SM is a challenging variable. However, Qiao et al. (2022) show that CMIP6 models reproduce the climatology and seasonal variability of SM reasonably well over SSA when compared with ERA5. As for the projected changes in SM ($\Delta$SM), Figure R5 demonstrates a substantial uncertainty in regional $\Delta$SM. Moreover, the spread

of ΔSM across the CMIP6 models is clearly distinguishable from the internal variability (Table R1). To further illustrate this, we compared the magnitude of the projected ΔSM with an estimate of the internal variability (Figure R8), following the approach in Mindlin et al. (2020) (appendix B). In the southernmost region (SS), all models project a robust drying, with confidence intervals not overlapping zero, meaning that the drying signal is robust relative to interannual variability, although the magnitude of the change is uncertain. Central-eastern regions (CA and CES) also exhibit a large spread, spanning drying, no significant changes or moistening. Similar results are obtained for the other drivers (see Table R2 and the new Table S2 in the revised manuscript). Consequently, the following text has been included in the revised version of the manuscript (lines 292-297):
*"For all drivers, the spread of responses across the multi-model ensemble is clearly distinguishable from the internal variability. To illustrate this, we compared the magnitude of the projected changes in the selected drivers with an estimate of the internal variability based on the interannual standard deviation of the detrended series, following the approach in Mindlin et al. (2020) (appendix B). The results (Table S2) show that the variability within individual models is significantly lower than across models (inter-model variability), indicating that the spread in driver projections is primarily driven by model uncertainty rather than by internal variability."*

[Figure]

*Figure R8.* Summer (DJF) soil moisture changes scaled by global warming (ΔSM; kg m$^{-2}$ K$^{-1}$) for each model in CES, CA, and SS regions. Error bars represent $\pm 1.96 \times$ SE, where SE is the standard deviation of the detrended SM series of the historical period, following Mindlin et al. (2020, Appendix B). Red dashed line indicates the multi-model mean of ΔSM.

**I am not convinced about how appropiate the index of local Z500 to be representative of model uncertainty in circulation change. Is Z500 particularly uncertain over the area where it is averaged? This driver is used to characterize changes around 50S. From a dynamics perspective, any local circulation driver in this region has to capture the uncertainty in regional circulation change. For example, as the author reference Mindlin et al. 2020, the changes there are understood in terms of changes in the westerlies, which influence this region (Figure 3b and 3f – if I can interpret this figure correctly, see below). What is the circulation**

**feature that is projected to change and is captured by this index? Is the climatological strength of Z500 locally a circulation feature that is chaning in the models?**

We understand the reviewer's concern. We acknowledge that the regional TXx anomalies are also influenced by the large-scale westerly circulation, which dominates at extratropical scales. However, in this study, the $Z500_{HL}*$ index was used primarily as a regional indicator of the westerlies. In our previous work (Suli et al., 2023), we found that temperature extremes, such as heatwaves, are mainly triggered by anticyclonic anomalies, whose origin and characteristics vary across SSA regions. For instance, in the southernmost region (SS), heatwaves are predominantly associated with high-latitude blocking events. In CES, CA and NS regions, heatwaves are closely linked to subtropical high-pressure systems over the oceans. Based on this evidence, we selected the $Z500_{HL}*$ index to capture the weakening of the westerlies (polar jet) by the presence of high-latitude Z500 departures with respect to the zonal mean (i.e. ridging). Positive values of this driver indicate a weakening of the polar westerlies linked to high-latitude blocking over SS, whereas negative values reflect a strengthened eddy-driven jet associated with mid-latitude intrusions of subtropical high-pressure systems from the ocean to the continent (Figure R8). Therefore, the index is designed to collectively capture the relevant circulation features of each region (high-latitude blocking, mid-latitude ridges, etc.).

Based on the reviewer's comment, we have added a new Figure in the supplement (Figure S1) and the following clarification to the revised manuscript (lines 158–162): *"This index is used as a proxy for regional ridging activity and associated intensity of the westerlies over SSA. Positive $Z500_{HL}*$ values indicate enhanced high-latitude blocking, whereas negative values reflect mid-latitude high-pressure systems (Figure S1), thus capturing the range of regional circulation patterns that favour extremely high temperatures across SSA (Suli et al. 2023)."*

[Figure]

*Figure R8. MMM Pearson correlation coefficients of the summer (DJF) series of the $Z500_{HL}*$ driver and Z500 during the historical period (1979–2014). The box indicates the region used to construct the driver and stippling areas denote regions where at least 66%*

*of the models show significant correlations (p < 0.1) of the same sign.*

To further address whether this driver is changing in the models, we computed $\Delta Z500_{HL}*$ for each model. More than 60% of the models (16/26) project a positive $\Delta Z500_{HL}*$, indicating an intensification of high-latitude ridging, although the spread across models is substantial. Indeed, the inter-model spread in $\Delta Z500_{HL}*$ is significantly larger than its internal variability (see Table R2 and the new Table S2 in the revised text). This supports the use of SSA ridging as a relevant circulation driver of model-dependent changes in TXx.

**215 I don't understand why the drivers cannot be correlated. Isn't the multi-linear regression framework used precisely used to controll for any confounding effect? Drivers can be correlated in multi-linear regression frameworks. I recommend the authors this review article on how to interpret regression models: http://dx.doi.org/10.2139/ssrn.3689437**

Thanks for the reference. While it is true that multi-linear regressions (MLRs) can estimate partial effects even with correlated predictors, high correlations reduce the clarity and stability of the MLR coefficients. In our study we seek to minimise the correlation between drivers to avoid redundant information that adds unnecessary complexity to the model, which can in turn increase the risk of overfitting. By ensuring that drivers are not correlated, each coefficient clearly reflects the distinct physical influence of its respective driver, which is essential for an unambiguous interpretation of the storylines across regions. Recall that for simplicity, we restrict the analysis to storylines derived from two drivers (four storylines). If the two drivers were strongly correlated, the storylines would effectively describe the influence of a single driver. This would hamper the potential of the storylines to sample uncertainty and uncover competing effects between independent drivers' changes. This has been clarified in the revised text as follows (lines 267-269): *"We also verified that the selected drivers were not significantly correlated with each other (i.e. Pearson correlation coefficients with p-values > 0.1) to avoid redundant information that adds unnecessary complexity to the model and interpretation of the drivers."*

**I don't understand the discussion related to uncertainty in TXx changes over CES. If it is explained by soil moisture, why are storylines based on this driver presented?**

We assume that the reviewer refers to the non-significant response of $\Delta$TXx to SM changes over the CES region. We have revised the text to clarify that each regional driver was identified based on the strength and physically consistency of its relationship with regional TXx. Then, we performed separate MLR models for each region. It is important to highlight that keeping the two drivers enables to construct the storyline analysis and assess their combined effects on $\Delta$TXx. Therefore, we include the storylines of $\Delta$TXx in CES to maintain a consistent framework across the analysed regions, even though one of the drivers shows a weaker statistical relationship. This has been stressed in the revised manuscript (lines 235-236): *"To identify remote and local drivers of $\Delta$TXx, we examined*

*the regression patterns of several variables and constructed indices displaying a strong and physically consistent relationship with regional TXx."*

**230 A correlation analysis is useful to show that the target variable is correlated with changes in the region. However, the way that the results are presented the caption is wrong, as well as some parts of the interpretation of the maps in the text. The authors say "*Figure 3 illustrates the corresponding sensitivity patterns, as obtained from a MLR of regional ΔTXx onto grid-point drivers' changes*" but the caption says: "*Sensitivities of summer changes in absolute maximum temperature (ΔTxx, 2070–2099 minus 255 1979–2014) associated with uncertainties in the responses of key drivers for each region determined using a multi linear regression model (see Eq. 1)*". These are not the same analysis. What does "*grid-point drivers' changes* " mean? I understood that the drivers were indices, not fields. If these are the regression coefficients of Eq 1 bx and cx, then the coeffcents show how much of Δ$TXx$/GW is explained by each of the drivers. If this is the case, I don't understand what the authors are trying to interpret from the map. For example, N3.4 is correlated with the temperatures in the region where the index is computed. This is clearly to be expected, but it has nothing to do with the temperatures in the study area. In any case, it would make sense if the authors correlated the changes Δ$TXx$/GW onto SST at each grid point , OLR at each grid point, etc. This is done in https://doi.org/10.1029/2023JD038712 to identify drivers of uncertainty in their study area.**

**In my understanding, the following statement "*Regarding CA (Figs. 3 e-f), the results show that GCMs with larger decreases in SMCA or more pronounced poleward shifts of the SACZ display exacerbated warming of TXx.*" is a correct interpretation of the maps if the statement in the caption is correct. But then the description of the figure "Figure 3 illustrates the corresponding sensitivity patterns, as obtained from a MLR of regional ΔTXx onto grid-point drivers' changes" is unclear.**

We regret the misunderstanding. By "grid-point drivers' changes" we meant the responses in the 2D fields from which the drivers are derived (e.g. SST for the ENSO driver). Our original intention was to use this figure to justify the selection of the drivers and illustrate their influence. However, as the comment noted, this could be confusing, so we updated the figure. It now shows the summer changes in each 2D variable (i.e., SST, OLR, Z500 or SM) for a scaled 1 K/K warming in regional maximum temperature (Δ$TXx$/GW). These regression maps are now included in Section 3.2 (Selection of drivers), and they are used to identify drivers of uncertainty for each SSA region. The boxes in each panel indicate the regions from which the driver indices were later computed for the MLR analysis. Both the text and figure caption have been revised to clarify this:

Revised manuscript (lines 236-237): *"Figure 3 illustrates the linear regression patterns of these field responses onto regional ΔTXx."*

Caption of Figure 3 (lines 257-261): *"Regression-based summer changes in several fields (expressed in SD with respect to the MMM) corresponding to a 1 K/K regional scaled warming (ΔTxx / GW, 2070–2099 minus 1979–2014) for: NS: (a) ΔSST and (b) ΔOLR; CES: (c) ΔSM and (d) ΔOLR; CA: (e) ΔSM and (f) ΔOLR; SS: (g) ΔSM and (h) ΔZ500.*

*Boxes indicate the regions used to construct regional driver indices for the MLR analysis. Local drivers are denoted in green and remote drivers in yellow. Stippling denotes statistically significant regression coefficients at p < 0.1, after a two-tailed t-test."*

**It would be relevant to present the robustness of the responses under the storylines. Could the authors provide a reference of which is the mean or median absolute error of the regression model? The coefficient of determination is presented, and this is useful to understand how much of the model spread is explain. However, the absolute value of the errors is a better estimation of how robust the storyline estimates are, if they are developed using the regression framework. With this I mean, if you try to reconstruct each model's response with the regression framework, how good is this estimate? I understand that this was not done in Zappa & Shepherd (2017) but since then, for example in Mindlin et al. (2020) who you cite, this was done. If errors are an order of magnitude smaller than the difference between storylines, one can ensure that using the regression model to represent the storylines is appropriate. It is important to show the robustness of the analysis, otherwise the conclusions can be based on statistical artifacts.**

Thanks for the suggestion. To provide a measure of the robustness of the responses under the storylines (STs), we calculated the median absolute error (MdAE) of the MLR for each SSA region (Table R3). The MdAE was then compared with the differences between opposite storylines of ΔTXx. In most regions (NS, CA and SS), the MdAE contributes only ~25% or less to the ST responses, which is near the lower bound (21%) of the range found by Mindlin et al. (2020). In the CES region, the MdAE represents a higher fraction of the differences between STs (~35%), which may be due to the lack of significance of one of the drivers. Overall, these results confirm that the regression-based framework provides a meaningful representation of the TXx responses across SSA, and that the derived STs are not dominated by statistical artefacts. The information from the MdAE has been included as a new column in Table 1, and the following text has been added to the revised version (lines 317-322): *"To measure the robustness of the storylines, we compared the difference between opposite storylines (Fig. 5) with the median absolute error (MdAE) of the MLR (Table 1) for each SSA region, similar to Mindlin et al. (2020). In most regions (NS, CA and SS), the MdAE represents less than ~25% of the storyline responses. In CES, the MdAE represents a higher fraction of the differences between storylines (~35%), which may be due to the lack of significance of one of the drivers. Overall, these results confirm that the regression-based framework provides a meaningful representation of the TXx responses across SSA."*

| Region | ST1 (Low d1 + Low d2) | ST2 (High d1 + Low d2) | ST3 (Low d1 + High d2) | ST4 (High d1 + High d2) | MdAE |
|---|---|---|---|---|---|
| NS | 0.95 | 1.16 | 1.24 | 1.46 | 0.14 |
| CES | 0.95 | 0.93 | 1.20 | 1.18 | 0.10 |
| CA | 1.08 | 0.86 | 1.24 | 1.02 | 0.06 |
| SS | 1.04 | 0.89 | 1.16 | 1.01 | 0.07 |

*Table R3.* Median absolute errors (MdAE) of the MLR for each SSA region, compared with the TXx responses obtained under the four storylines (ST1–ST4). Units are K/K

**References**

Cinelli, Carlos and Forney, Andrew and Pearl, Judea, A Crash Course in Good and Bad Controls (September 9, 2020). http://dx.doi.org/10.2139/ssrn.3689437

Monerie, P.-A., Biasutti, M., Mignot, J., Mohino, E., Pohl, B., & Zappa, G. (2023). Storylines of Sahel precipitation change: Roles of the North Atlantic and Euro-Mediterranean temperature. Journal of Geophysical Research: Atmospheres, 128, e2023JD038712. https://doi.org/10.1029/2023JD038712

Narsey, S., J. R. Brown, F. Delage, G. Boschat, M. Grose, R. Colman, and S. Power, 2022: Storylines of South Pacific Convergence Zone Changes in a Warmer World. *J. Climate*, 35, 6549–6567, https://doi.org/10.1175/JCLI-D-21-0433.1.